# Label GM-PHD Filter Based on Threshold Separation Clustering

**DOI:** 10.3390/s22010070

**Published:** 2021-12-23

**Authors:** Kuiwu Wang, Qin Zhang, Xiaolong Hu

**Affiliations:** 1School of Air Defense and Missile Defense, Air Force Engineering University, Xi’an 710051, China; kinzh@263.net (Q.Z.); huxiaolong0@126.com (X.H.); 2Graduate School, Air Force Engineering University, Xi’an 710051, China

**Keywords:** multi-target tracking (MTT), random finite set (RFS), probability hypothesis density filter, gaussian mixture, label track maintenance

## Abstract

Gaussian mixture probability hypothesis density (GM-PHD) filtering based on random finite set (RFS) is an effective method to deal with multi-target tracking (MTT). However, the traditional GM-PHD filter cannot form a continuous track in the tracking process, and it is easy to produce a large number of redundant invalid likelihood functions in a dense clutter environment, which reduces the computational efficiency and affects the update result of target probability hypothesis density, resulting in excessive tracking error. Therefore, based on the GM-PHD filter framework, the target state space is extended to a higher dimension. By adding a label set, each Gaussian component is assigned a label, and the label is merged in the pruning and merging step to increase the merging threshold to reduce the Gaussian component generated by dense clutter update, which reduces the computation in the next prediction and update. After pruning and merging, the Gaussian components are further clustered and optimized by threshold separation clustering, thus as to improve the tracking performance of the filter and finally realizing the accurate formation of multi-target tracks in a dense clutter environment. Simulation results show that the proposed algorithm can form a continuous and reliable track in dense clutter environment and has good tracking performance and computational efficiency.

## 1. Introduction

Multi-target tracking (MTT) is a process of assigning the measured values to the targets, filtering them at the same time, and managing the tracks of multiple targets according to the time step [1,2]. RFS provides a unified and clear framework for MTT. Probability Hypothesis Density (PHD) [3] filtering has a good potential in solving target tracking under insufficient prior knowledge and an unknown number of targets. At present, it has been widely used in radar target tracking [4,5,6], computer vision [7], real-time positioning and map building [8], and group target tracking [9].

Traditional tracking algorithms, such as probabilistic data association (PDA) and multiple hypothesis tracking (MHT) [10,11,12,13], all transform the multi-target problem into a parallel single-target tracking problem by assigning measurements. The core of its processing method is data correlation, but when there are many targets and a large number of false alarm clutter, the correlation will bring a combination explosion and make the calculation amount increase exponentially. Correlation error and state estimation error are coupled and influence each other, resulting in large estimation error. PHD filtering technology avoids complex data association problems in the process of state estimation and can concentrate resources to deal with tracking problems. Thus, it has good application performance. Among them, Gaussian mixture (GM) and sequential Monte Carlo (SMC) are two important methods of PHD operation, which are called GM-PHD [14] and SMC-PHD [15], respectively. For the SMC single target algorithm, reference [16] proposed an adaptive model discrimination method, and does not need to be adjusted for specific problems, and has a better effect on the model uncertainty of discrete data. Reference [17] does not use model selection but integrates model information in the SMC method and dynamically adjusts the sampling step through the posterior predictive ability of each model to adapt to the uncertain environment of the model. Reference [18] uses interactive parallel particle filters, each filter is for a different model, and the method is simple and flexible, which solves the joint problem of online tracking and detection of the current modality. For the GM multi-target algorithm, reference [19] proposed an improved probability hypothesis density filter to modify the weight of the target when the target was missed. Reference [20] proposed a modified GM-PHD filter, which corrected the prediction and update equation of the PHD filter, and reduced the problem of information loss caused by a missed detection. Reference [21] improved the filter performance in the case of continuously missed alarms by establishing a survival probability model and adaptively adjusting the confirmation probability. This article is researched on the basis of GM-PHD.

PHD filter has a rigorous mathematical theoretical basis, which can realize joint detection and tracking of targets. The structure is complete and clear, and the amount of calculation is small. PHD filter cannot identify the tracks of different targets in the tracking process. With the increase of targets or the approach of distances, the wrong judgment of tracks leads to the inability of the GM-PHD filter to track the targets that need attention. Therefore, for a PHD filter, the accurate distinction of tracks is the key to ensuring tracking performance [22,23,24].

In MTT, it is the premise of establishing the track on determining the identity of the target. The data association algorithm [25] proposed by Panta provided a unique label for each target and obtained the track of a single target through the state at each time and the association between the targets [14]. Vo et al. [26,27,28] also proposed label Multi-Bernoulli filter and label multi-target Bayesian processing method, which effectively solved the track formation problem of RFS processing multi-target, and belonged to the pioneering work of label RFS. Reference [29] used dynamic weight allocation based on detection to reduce the weight error of adjacent targets and proposed a novel tracking management scheme, which effectively solved the track recognition of adjacent targets [30]. In addition, Reference [30] proposed a track extraction method based on target topology information to identify targets and manage tracks.

In the background of dense clutter, the PHD filter is difficult to form a continuous and stable track and has a large tracking error. In the framework of GM-PHD filtering, this paper introduces the Gaussian component label set into the pruning and merging step and proposes a threshold separation clustering algorithm considering velocity and position information to extract the target state. This method not only outputs the target track accurately but also improves the tracking accuracy and reduces the computational complexity. The main contributions of this paper are as follows: (1) the merging and pruning step is improved, the Gaussian components with the same label are merged, and the continuous missing detection parameters are used for moderate track preservation. (2) A method of clustering Gaussian components by threshold separation is proposed. After pruning and merging, Gaussian components are clustered by velocity threshold and Euclidean distance. The filtering performance of the improved algorithm has been verified by simulation scenes.

The following chapters are arranged as follows: Section 2 describes the traditional PHD filtering, including PHD recursion, Gaussian mixture implementation, and label GM-PHD algorithm. Section 3 introduces the improvement of label GM-PHD filtering, including the merging and pruning steps of introducing label components, threshold separation clustering method, the formation of tracking associated tracks, and the comprehensive implementation of the algorithm. In Section 4, the effectiveness of the algorithm is verified by linear scene simulation experiments. Section 5 is the conclusion.

## 2. PHD Filtering

### 2.1. PHD Recursion

Considering the following MTT scenario, the target state set is $X_{k}=\left\{x_{k}^{1},\cdots ,x_{k}^{N_{k}}\right\}$ and the measurement set is $Z_{k}=\left\{z_{k}^{1},\cdots ,z_{k}^{M_{k}}\right\}$, where $x_{k}^{n}$ and $z_{k}^{m}$ represent the *n*-th target state and the *m*-th measurement at time *k*, respectively. $N_{k}$ and $M_{k}$ are the number of targets and the number of measurements at time *k*, respectively, and they change with time and the environment. Assuming that the prior probability of multi-target approximately obeys Poisson distribution, according to the RFS statistics theory, the PHD recurrence equation is [3]
(1)$$\begin{array}{ll} D_{k|k-1}\left(x\right)= & \int \left(p_{s,k|k-1}f_{k|k-1}\left(x|\zeta \right)\right)+\beta_{k|k-1}\left(x|\zeta \right)D_{k-1|k-1}\left(\zeta \right)d\zeta \\ & +\gamma_{k}\left(x\right)\; \end{array}$$
(2)$$\begin{array}{l} D_{k|k}\left(x\right)=\left[1-P_{D,k}\right]D_{k|k-1}\left(x\right)\\ +\sum_{z_{k}\in Z_{k}}\frac{P_{D,k}g_{k}\left(z|x\right)D_{k|k-1}\left(x\right)}{\lambda c\left(z_{k}\right)+\int P_{D,k}g_{k}\left(z|\zeta \right)D_{k|k-1}\left(\zeta \right)d\zeta } \end{array}$$$\gamma_{k}\left(x\right)$ and $\beta_{k|k-1}\left(x|\zeta \right)$ represent the target strength of newborn and spawned RFS respectively [31], $p_{s,k|k-1}$ is the survival probability of target at time *k* on *k −* 1, $P_{D,k}$ represents the detection probability of target at time *k*, $f_{k|k-1}\left(x|\zeta \right)$ is the probability density function of state transition, $g_{k}\left(z|x\right)$ is the likelihood function of a single target, and λ is the average number of clutter, and $c\left(z_{k}\right)$ is probability density functions of Poisson clutter distribution.

### 2.2. Gaussian Mixture Implementation

GM-PHD filter represents the prior PHD (1) and the posterior PHD (2) of multi-target as Gaussian mixture formation, and its iterative recursion can be expressed as a prediction update structure similar to Kalman filter (KF), and the algorithm is processed in a multi-target space.

Assume that the multi-target posterior PHD at time *k −* 1 can be expressed as a Gaussian mixture, and the equation is as follows:(3)$$D_{k-1}\left(x\right)=\sum_{i=1}^{J_{k-1}}w_{k-1}^{i}N\left(x;m_{k-1}^{i},p_{k-1}^{i}\right)$$
where $J_{k-1}$ is the number of Gaussian components at time *k −* 1. In general, the number of $J_{k}$ is much greater than the number of $N_{k}$. $w_{k-1}^{i}$ is the weight of the *i*-th Gaussian component, $m_{k-1}^{i}$ and $p_{k-1}^{i}$ represents the mean and covariance of the *i*-th Gaussian component. Then time *k* multi-target prediction PHD and time *k* posterior multi-target PHD can be expressed as Gaussian mixture form:(4)$$D_{k|k-1}\left(x\right)=\gamma_{k}\left(x\right)+D_{\beta_{k|k-1}}\left(x\right)\;+D_{sv_{k|k-1}}\left(x\right)\;\;=\sum_{i=1}^{J_{k|k-1}}w_{k|k-1}^{i}N\left(x;m_{k|k-1}^{i},p_{k|k-1}^{i}\right)$$
(5)$$m_{k|k-1}^{i}=Fm_{k-1}^{i}$$
(6)$$p_{k|k-1}^{i}=Q_{k}+Fp_{k-1}^{i}{\left(F\right)}^{T}$$
(7)$$\begin{array}{l} D_{k}\left(x\right)=\left[1-P_{D,k}\right]D_{k|k-1}\left(x\right)\\ +\sum_{l=1}^{m_{k}}\sum_{i=1}^{J_{k|k-1}}w_{k}^{j}\left(z\right)N\left(x;m_{k}^{i},p_{k}^{i}\right) \end{array}$$
(8)$$w_{k}^{i}\left(z\right)=\frac{P_{D,k}w_{k|k-1}^{i}g_{k|k-1}\left(z|m_{k|k-1}^{i},p_{k|k-1}^{i}\right)}{\lambda c\left(z_{k}^{l}\right)+P_{D,k}\sum_{j=1}^{J_{k|k-1}}w_{k|k-1}^{i}g_{k|k-1}\left(z|m_{k|k-1}^{j},p_{k|k-1}^{i}\right)}$$
(9)$$K_{k}^{i}={\hat{p}}_{k|k-1}^{i}H_{k}^{T}{\left[H_{k}{\hat{p}}_{k|k-1}^{i}H_{k}^{T}+R_{k}\right]}^{-1}$$
(10)$${\hat{m}}_{k}^{i}={\hat{m}}_{k|k-1}^{i}+K_{k}^{i}\left(z_{k}^{l}-H_{k}{\hat{m}}_{k|k-1}^{i}\right)$$
(11)$${\hat{p}}_{k|k}^{i}=\left[I-K_{k}^{i}H_{k}\right]{\hat{p}}_{k|k-1}^{i}$$
where $D_{\beta_{k|k-1}}\left(x\right)$ and $D_{sv_{k|k-1}}\left(x\right)$ are the Gaussian mixture intensity function of spawned target and surviving target respectively, $J_{k|k-1}$ is the number of Gaussian components of predicting PHD, and $w_{k|k-1}^{i}$, $m_{k|k-1}^{i}$ and $p_{k|k-1}^{i}$ are respectively the weight, mean, and covariance of the i-th Gaussian component in the prediction intensity function. $w_{k}^{j}$, $m_{k}^{j}$, and $p_{k}^{j}$ are, respectively, the weight, mean, and covariance of the *j*-th component in the update intensity function, F is the state transition matrix, $H_{k}$ is the measurement transition matrix, $Q_{k}$ is the process noise covariance, $R_{k}$ is the measurement noise covariance matrix, and *I* is the identity matrix.

The expected number of targets ${\hat{N}}_{k|k-1}$ and ${\hat{N}}_{k}$ associated with $D_{k|k-1}$ and $D_{k}$ are obtained as follows:(12)$${\hat{N}}_{k|k-1}={\hat{N}}_{k-1}\left(p_{s,k}+\sum_{j=1}^{J_{\beta ,k}}w_{\beta ,k}^{j}\right)+\sum_{j=1}^{J_{\gamma ,k}}w_{\gamma ,k}^{j}$$
(13)$${\hat{N}}_{k}={\hat{N}}_{k|k-1}\left(1-p_{D,k}\right)+\sum_{z\in Z_{k}}\sum_{j=1}^{J_{k|k-1}}w_{k}^{j}\left(z\right)$$

Among them, $J_{\beta ,k}$ and $J_{\gamma ,k}$, respectively, represent the number of spawned Gaussian components and newborn Gaussian components at time *k*.

Therefore, the terms required to represent $D_{k}$ at step k is $J_{k}=\left(J_{k-1}\left(1+J_{\beta ,k}\right)+J_{\gamma ,k}\right)\left(1+\left|Z_{k}\right|\right)$. However, the number of Gaussian components and the computational complexity increases very fast.

### 2.3. Label GM-PHD Filter

In Reference [14], a tracking label was introduced into the GM-PHD filter, which extracts the target track by assigning a label to each Gaussian component. Its algorithm includes the following steps:

Step 1: initialize the Gaussian mixture posterior PHD at t=0:(14)$$D_{0}\left(x\right)=\sum_{i=1}^{J_{0}}w_{0}^{i}N\left(x;m_{0}^{i},p_{0}^{i}\right)$$

Assign a unique label to each Gaussian component:(15)$$\Gamma_{0}=\left\{\tau_{0}^{1},\cdots ,\tau_{0}^{J_{o}}\right\}$$$\tau_{0}^{i}$ is label of the *i*-th Gaussian component, and the mean is $m_{0}^{i}$ and covariance is $p_{0}^{i}$.

Step 2: Predict according to Equation (4), the label set $\left(\Gamma_{k|k-1}\right)$ construction method is as follows:

The surviving Gaussian component still retains the original label, and the newborn or spawned Gaussian component assigns a newborn label, then the predicted label set is defined as:(16)$$\Gamma_{k|k-1}=\Gamma_{k-1}\cup \left\{\tau_{\beta ,k}^{1,1},\cdots ,\tau_{\beta ,k}^{J_{k-1},J_{\beta ,k}}\right\}\cup \left\{\tau_{\gamma ,k}^{1},\cdots ,\tau_{\gamma ,k}^{J_{\gamma ,k}}\right\}$$

Among them, $\tau_{\beta }$ is a spawned Gaussian component label and $\tau_{\gamma }$ is a newborn Gaussian component label.

Step 3: Updating is performed according to Equation (7), and all Gaussian terms obtained after updating are assigned the same labels as their previously associated prediction terms.

Step 4: Pruning and merging

${\left\{\omega_{k}^{i},m_{k}^{i},p_{k}^{i}\right\}}_{i=1}^{J_{k}}$ is the PHD Gaussian component set updated at time *k*; Set a truncation threshold $T_{r}$, a merging threshold $T_{m}$, a maximum number of Gaussian terms $J_{\max}$, and two sets *A* and *B*.

Gaussian components weighted greater than the truncation threshold are put into set *A*:(17)$$A=\left\{i\in \left\{1,\cdots ,J_{\max}\right\}|\omega_{k}^{i}>T_{r}\right\}$$

Then find out the index value of Gaussian term with the largest weight in set *A*: $j=\arg\underset{i\in A}{\max}\omega_{k}^{i}$, and make set *B* as
(18)$$B=\left\{i\in A|{\left(m_{k}^{i}-m_{k}^{j}\right)}^{T}{\left(p_{k}^{i}\right)}^{-1}\left(m_{k}^{i}-m_{k}^{j}\right)\leq T_{m}\right\}$$

If the two Gaussian components satisfy Equation (18), they are combined into one term. The combined component parameters are obtained by weighted summation:(19)$${\tilde{\omega }}_{k}^{c}=\sum_{i\in B}\omega_{k}^{i}$$
(20)$${\tilde{m}}_{k}^{c}=\frac{1}{{\tilde{\omega }}_{k}^{c}}\sum_{i\in B}\omega_{k}^{i}m_{k}^{i}$$
(21)$${\tilde{p}}_{k}^{c}=\frac{1}{{\tilde{\omega }}_{k}^{c}}\sum_{i\in B}\omega_{k}^{i}\left(p_{k}^{i}+\left({\tilde{m}}_{k}^{c}-m_{k}^{i}\right){\left({\tilde{m}}_{k}^{c}-m_{k}^{i}\right)}^{T}\right)$$

Step 5: Multi-target state extraction

Besides extracting the target state, the set composed of Gaussian component labels is also extracted
(22)$${\hat{\Gamma }}_{k}=\left\{\tau_{k}^{i}|\omega_{k}^{i}>0.5\right\}$$
(23)$${\hat{X}}_{k}=\left\{m_{k}^{i}|\omega_{k}^{j}>0.5\right\}$$

Step 6: Track extraction

At each moment, if there is a Gaussian component or its spawned Gaussian term $\omega_{k}^{i}>0.5$, it is marked as “confirmed” and other branches at $\omega_{k}^{i}\leq 0.5$ are marked as “tentative”.

## 3. Robust Label GM-PHD Filter

In a dense clutter environment, due to the existence of a large number of clutters, a large number of Gaussian components with unknown sources will appear in the pruning and merging step. If not properly processed, the computational complexity of the label GM-PHD filter will be greatly increased in the subsequent prediction and update steps. The existence of dense clutter will also affect the final state extraction of the filter and then directly affect the formation of correct multi-target tracks. It is necessary to consider improving the pruning and merging method, and on this basis, optimize the state extraction process of the algorithm, thus that the output track is correct and stable, and at the same time improve the efficiency of the algorithm.

### 3.1. Improve Pruning and Merging Methods

As mentioned earlier, GM-PHD filter uses the Gaussian mixture distribution to model multi-target to update the posterior strength of multi-target. However, such as other PHD filters, this filter does not consider the time sequence correspondence of Gaussian components in the tracking process. Thus, it is impossible to distinguish targets. Label GM-PHD filtering method solves this problem, which uses pruning and merging process to exclude unreal targets and ensure the formation of tracks. However, the pruning and merging method does not consider the label set of Gaussian components. When the target was occluded, or the distance was close, the Gaussian components corresponding to the real target were pruned or merged, which significantly degrades the tracking performance of the algorithm. The main reason why targets cannot be tracked when they are close to each other is that the label of the Gaussian component is ignored. If the merging threshold $T_{m}$ is set relatively large, the number of merged Gaussian components will increase, which will lead to an increase in the probability of mixing different labels, thus that the 2 targets cannot be effectively distinguished. If the merge threshold setting $T_{m}$ is small, it is not conducive to the aggregation of real targets. A large number of invalid Gaussian components are still retained, which greatly increases the subsequent calculation burden. The above problems can be solved by the following improvements, that is, only Gaussian terms with the same label were allowed to be merged. This processing was not only beneficial to the problem of multi-target discrimination in dense clutter environments but can also consider setting a larger merging threshold $T_{m}$ to improve the target tracking performance. The expression is as follows:(24)$$B=\left\{i\in A|{\left(m_{k}^{i}-m_{k}^{j}\right)}^{T}{\left(p_{k}^{i}\right)}^{-1}\left(m_{k}^{i}-m_{k}^{j}\right)\leq T_{m},\tau_{k}^{i}=\tau_{k}^{j}\right\}$$

The combined component parameters were obtained by Equations (19)–(21). In order to reduce the operation time of each step, reference [31] proposes a tree structure, in which the initial Gaussian component was set as the origin, and if there were Gaussian components or their spawned Gaussian terms $\omega_{k}^{i}>0.5$, the tree was marked as “confirmed”, and other branches unsatisfied the $\omega_{k}^{i}>0.5$ were marked as “tentative”. In the process of target tracking, due to the influence of clutter and noise, the target may disappear for several frames before appearing. The confirmed Gaussian component $n_{missed}$ was assigned a parameter representing the number of consecutive missed detections on the branch. The method allows previously confirmed components $n_{missed}\leq 3$ marked as tentative in the current step.

The proposed algorithm flow is as follows:
1. For ${\left\{\omega_{k}^{i},m_{k}^{i},p_{k-1}^{i}\right\}}_{i=1}^{J_{k}}$, the truncation threshold $T_{r}$, the merging threshold $T_{m}$ and the maximum number of Gaussian terms $J_{\max}$2. Set the set $A=\left\{i\in \left\{1,\cdots ,J_{\max}\right\}|\omega_{k}^{i}>T_{r}\right\}$3. Find the index value of the maximum weight value in set *A*
$$j=\arg\underset{i\in A}{\max}\omega_{k}^{i}$$

Obtain the set $B=\left\{i\in A|{\left(m_{k}^{i}-m_{k}^{j}\right)}^{T}{\left(p_{k}^{i}\right)}^{-1}\left(m_{k}^{i}-m_{k}^{j}\right)\leq T_{m},\tau_{k}^{i}=\tau_{k}^{j}\right\}$



4. Weighted summation to obtain Gaussian component parameters
$${\tilde{\omega }}_{k}^{c}=\sum_{i\in B}\omega_{k}^{i}$$
$${\tilde{m}}_{k}^{c}=\frac{1}{{\tilde{\omega }}_{k}^{c}}\sum_{i\in B}\omega_{k}^{i}m_{k}^{i}$$
$${\tilde{p}}_{k}^{c}=\frac{1}{{\tilde{\omega }}_{k}^{c}}\sum_{i\in B}\omega_{k}^{i}\left(p_{k}^{i}+\left({\tilde{m}}_{k}^{c}-m_{k}^{i}\right){\left({\tilde{m}}_{k}^{c}-m_{k}^{i}\right)}^{T}\right)$$5. Put component parameter $\left\{{\tilde{\omega }}_{k}^{c},{\tilde{m}}_{k}^{c},{\tilde{p}}_{k}^{c}\right\}$ into the output $Output_{k}$
$$Output_{k}=\left\{Output_{k},\left\{{\tilde{\omega }}_{k}^{c},{\tilde{m}}_{k}^{c},{\tilde{p}}_{k}^{c}\right\}\right\}$$


### 3.2. Threshold Separation Clustering

Based on the improve pruning and merging methods, GM-PHD extracts Gaussian components with weight greater than 0.5 after pruning and merging as output and retains the remaining Gaussian components to continue iterative filtering. Due to the persistent interference of strong clutter in a dense clutter environment, it is easy to produce wrong Gaussian terms with large weights. If they are extracted and regarded as newborn targets, it will lead to high tracking errors.

It can be seen that finding the real target in a dense clutter environment and effectively separating the real target from the wrong Gaussian term are the key to solving the problem. Reference [32] proposed a threshold separation clustering method. On this basis, this paper proposes a method of clustering Gaussian components of the same label using velocity and distance information and extracting the target state, which can effectively aggregate the real target state and prevent redundant error. The Gaussian term interferes with the extraction of the target state and can reduce the amount of calculation on the basis of [32].

Firstly, the index value of the Gaussian component with the largest weight in set *B* was found, and the weight value was placed in the set $S_{\omega }$.
(25)$$j=\arg\;\underset{n=1:N}{\max}\left(\omega_{k}^{n}\right)$$
(26)$$S_{\omega }=S_{\omega }\cup \left\{\omega_{k}^{j}\right\}$$
where *N* represents the number in set *B* after pruning and merging.

The velocity difference and the corresponding Euclidean distance between the Gaussian component with the largest weight and the Gaussian component with the same label in set *B* were calculated.
(27)$$speed_{i,j}=\left(\left(m_{k}^{j}-m_{k}^{i}\right)*R_{0}\right){\left(p_{k}^{j}\right)}^{-1}{\left(\left(m_{k}^{j}-m_{k}^{i}\right)*R_{0}\right)}^{T},\tau_{k}^{i}=\tau_{k}^{j}$$
(28)$$d_{i,j}=\sqrt{{\left(m_{k}^{j}*R_{1}-m_{k}^{j}*R_{2}\right)}^{2}+{\left(m_{k}^{i}*R_{1}-m_{k}^{i}*R_{2}\right)}^{2}},\tau_{k}^{i}=\tau_{k}^{j}$$
wherein $R_{0}$ is used for the velocity information for separating, $R_{1}$ and $R_{2}$ is used for the position information for separating the target. In a linear system, $R_{0}$ is column matrices ${\left[0\;1\;0\;1\right]}^{T}$, and $R_{1}$ and $R_{2}$ are column matrices ${\left[1\;0\;0\;0\right]}^{T}$ and ${\left[0\;0\;1\;0\right]}^{T}$, respectively.

Finally, all Gaussian components satisfying the given velocity threshold $M_{speed}$ and Euclidean distance $M_{d}$ are clustered.
(29)$$S_{g,k}=\left\{x_{k}^{i}\in B|speed_{i,j}\leq M_{speed}\&\&d_{i,j}\leq M_{d}\right\}$$
(30)$$S_{k}=S_{k}\cup \left\{S_{g,k}\right\}$$
wherein the set $S_{k}$ contains all Gaussian components satisfying the conditions of Equations (27) and (28), and the clustering result is used as the final target state estimation.

### 3.3. Tracking Correlation to Form a Track

In the Gaussian component set $S_{k}$ obtained by threshold separation clustering, the Gaussian component with the largest weight in the corresponding label was found, which can effectively estimate the target track and introduce a track set ${ℑ}_{k}$.
(31)$$\zeta_{k}^{i}=\left\{r|r_{k}^{i}\in \tau_{k}^{i}\right\}$$
(32)$${ℑ}_{k}=\left\{\zeta_{k}^{i}\right\}\cup {ℑ}_{k-1}$$
wherein $\zeta_{k}^{i}$ is the i-th track corresponding to time *k*, and $r_{k}^{i}$ represents the corresponding track labeled $\tau_{k}^{i}$. If there is a track with a label $\tau_{k}^{i}$ in the set ${ℑ}_{k-1}$ and its parameter $n_{missed}=0$, $m_{k}^{i}$ would be added to the labeled $\tau_{k}^{i}$ track and added to the track set. This Gaussian component is confirmed to match the tracking label $\tau_{k}^{i}$.

If there is a track in the set ${ℑ}_{k-1}$ and has a label $\tau_{k}^{i}$, but it is not added to the track set ${ℑ}_{k}$, the number of missed detections is judged:(33)$$\left\{ \begin{array}{l} \zeta_{k}^{i}=\left\{r|r_{k}^{i}\in \tau_{k}^{i},n_{missed}^{i}\leq 3\right\}\\ m_{k}^{\left(i\right)}=F_{k}m_{k|k-1}^{i} \end{array}\right.$$

If Gaussian component $n_{missed}^{i}\leq 3$, $m_{k}^{i}$ would be added to the labeled track $\tau_{k}^{i}$.

Finally, multi-target state estimation can be obtained:(34)$${\hat{X}}_{k}=\left\{m_{k}^{i}:\tau_{k}^{i}\in {\hat{\Gamma }}_{k},i=1,\cdots ,N\right\}$$

### 3.4. Overall Steps to Improve the Algorithm

Aiming at the problem of poor tracking performance and low computational efficiency of traditional label GM-PHD in dense clutter environment, a robust label GM-PHD algorithm was proposed. The overall steps were as follows:

Step 1: Initialize the parameters

When t=0, a Gaussian mixture posterior PHD and an initial label are set, such as Equations (35) and (36):(35)$$D_{0}\left(x\right)=\sum_{i=1}^{J_{0}}w_{0}^{i}N\left(x;m_{0}^{i},p_{0}^{i}\right)$$
(36)$$\Gamma_{0}=\left\{\tau_{0}^{1},\cdots ,\tau_{0}^{J_{o}}\right\}$$

Each Gaussian component is recursive by the Bayesian criterion.

Step 2: Predict step

The Gaussian component is predicted according to Equation (4), and then newborn labels are assigned to the newborn and spawned Gaussian components, and the newborn label set and the spawned label set are added to the predicted label set:(37)$$\Gamma_{k|k-1}=\Gamma_{k-1}\cup \left\{\tau_{\beta ,k}^{1,1},\cdots ,\tau_{\beta ,k}^{J_{k-1},J_{\beta ,k}}\right\}\cup \left\{\tau_{\gamma ,k}^{1},\cdots ,\tau_{\gamma ,k}^{J_{\gamma ,k}}\right\}$$

Step 3: Update the steps

According to Equation (7), the prediction state is updated using the measurements at time *k*, and then all updated Gaussian terms are assigned the same label as their previously associated prediction terms.

Step 4: Pruning and merging

A state-updated Gaussian mixture PHD ${\left\{\omega_{k}^{i},m_{k}^{i},p_{k}^{i}\right\}}_{i=1}^{J_{k}}$ is obtained; set a truncation threshold $T_{r}$, a merging threshold $T_{m}$, a maximum number $J_{\max}$ of Gaussian terms, and two sets *A* and *B*.
(38)$$A=\left\{i\in \left\{1,\cdots ,J_{\max}\right\}|\omega_{k}^{i}>T_{r}\right\}$$

The Gaussian component satisfying the truncation threshold was found and put into the set *A*, and then the Gaussian term with the largest weight $j=\arg\max_{i\in A}\omega_{k}^{\left(i\right)}$ in the set *A* was found, and the Gaussian term was used for screening and merging:(39)$$B=\left\{i\in A|{\left(m_{k}^{i}-m_{k}^{j}\right)}^{T}{\left(p_{k}^{i}\right)}^{-1}\left(m_{k}^{i}-m_{k}^{j}\right)\leq T_{m},\tau_{k}^{i}=\tau_{k}^{j}\right\}$$

The combined component parameters were obtained by weighted summation:(40)$${\tilde{\omega }}_{k}^{c}=\sum_{i\in B}\omega_{k}^{i}$$
(41)$${\tilde{m}}_{k}^{c}=\frac{1}{{\tilde{\omega }}_{k}^{c}}\sum_{i\in B}\omega_{k}^{i}m_{k}^{i}$$
(42)$${\tilde{p}}_{k}^{c}=\frac{1}{{\tilde{\omega }}_{k}^{c}}\sum_{i\in B}\omega_{k}^{i}\left(p_{k}^{i}+\left({\tilde{m}}_{k}^{c}-m_{k}^{i}\right){\left({\tilde{m}}_{k}^{c}-m_{k}^{i}\right)}^{T}\right)$$

The Gaussian component of Set *A* that occurs in both Set *A* and Set *B* was deleted until Set *A* was an empty set.

Step 5: Threshold separation clustering

A velocity threshold and an Euclidean distance threshold were set, and Gaussian components satisfying Equations (43) and (44) and having the same label were clustered, and then Gaussian components satisfying conditions were put into a set $S_{k}$, and the clustering criteria were as follows:(43)$${\mathrm{speed}}_{i,j}=\left(\left(m_{k}^{j}-m_{k}^{i}\right)\times R_{0}\right){\left(p_{k}^{j}\right)}^{-1}{\left(\left(m_{k}^{j}-m_{k}^{i}\right)\times R_{0}\right)}^{T}$$
(44)$$d_{i,j}=\sqrt{{\left(m_{k}^{j}\times R_{1}-m_{k}^{j}\times R_{2}\right)}^{2}+{\left(m_{k}^{\left(i\right)}\times R_{1}-m_{k}^{i}\times R_{2}\right)}^{2}}$$

Step 6: Tracking association to form track

If there is a track with a label $\tau_{k}^{i}$ in the set ${ℑ}_{k-1}$ and its parameters $n_{missed}=0$, the corresponding Gaussian term $m_{k}^{i}$ is added to the labeled track $\tau_{k}^{i}$, and it is added to the track set ${ℑ}_{k}$, at which time the Gaussian component is confirmed to match the tracking label $\tau_{k}^{i}$.

If there is a track in the set ${ℑ}_{k-1}$ with a label $\tau_{k}^{i}$, but it is not added to the track set ${ℑ}_{k}$, the number of missed detections should be judged:(45)$$\left\{ \begin{array}{l} \zeta_{k}^{i}=\left\{r|r_{k}^{i}\in \tau_{k}^{i},n_{missed}^{i}\leq 3\right\}\\ m_{k}^{i}=F_{k}m_{k|k-1}^{i} \end{array}\right.$$

If Gaussian component $n_{missed}^{i}\leq 3$, $m_{k}^{\left(i\right)}$ would be added to the labeled track $\tau_{k}^{i}$.

## 4. Simulation and Experimental Results

### 4.1. Experimental Parameter Setting

In order to verify the effectiveness of the proposed algorithm in MTT scenarios, the improved label GM-PHD filter was compared with GM-PHD filter, label GM-PHD filter, and Gaussian mixture cardinalized probability hypothesis density (GM-CPHD) [28] filter. The experimental parameters were set as follows.

The tracking scene was set to multiple targets in four possible locations or spawned from other targets, and the observation area was $\left[-1000,1000\right]\times \left[-1000,1000\right]\left(m^{2}\right)$. There were 12 batches of targets in the scene. For the sake of simplicity, it was assumed that each target moves in a straight line at a uniform speed.

The state vector of the target consists of position and velocity components: $x_{k}=\left[p_{x,k}\;p_{y,k}{\;v}_{x,k}{\;v}_{y,k}\right]$, and its state equation is:(46)$$x_{k}=\left[\begin{array}{cccc} 1 & 0 & T & 0\\ 0 & 1 & 0 & T\\ 0 & 0 & 1 & 0\\ 0 & 0 & 0 & 1 \end{array}\right]x_{k-1}+\left[\begin{array}{cc} T^{2}/2 & 0\\ \begin{array}{l} 0\\ T \end{array} & \begin{array}{l} T^{2}/2\\ 0\\ T \end{array} \end{array}\right]w_{k-1}$$

The sampling interval *T* is 1 s, the total tracking time was 100 s, and the noisy process is $w_{k}∼N\left(0,5\right)$. The intensity of newborn target was as follows (47)
(47)$$\gamma_{k}\left(x\right)=\sum_{i=1}^{4}\omega_{\gamma ,k}^{i}N\left(x;m_{\gamma ,k}^{i},p_{\gamma ,k}^{i}\right)$$

Among them, $m_{\gamma ,k}^{1}={\left[400,-10,-600,5\right]}^{T}$, $m_{\gamma ,k}^{2}={\left[-800,20,-200,-5\right]}^{T}$, $m_{\gamma ,k}^{3}={\left[-200,15,800,-5\right]}^{T}$, $m_{\gamma ,k}^{4}={\left[0,-20,0,-15\right]}^{T}$, and weight of newborn target $\omega_{\gamma ,k}^{i}=0.03$. The process noise of newborn targets obeys Gaussian distribution, the mean value was zero, and the covariance was: $Q_{sp,k}^{i}=diag\left(\left[100,100,100,100\right]\right)$.

The measurement vector is position information: $z_{k}=\left[p_{z_{x},k}\;p_{z_{y},k}\right]$, and the measurement equation is given by the following equation:(48)$$z_{k}=\left[\begin{array}{cccc} 1 & 0 & 0 & 0\\ 0 & 1 & 0 & 0 \end{array}\right]x_{k}+v_{k}$$
where measured noise was $v_{k}∼N\left(0,5\right)$. Clutter is evenly distributed in the moving area of the target, with average scanning λ. In the Gaussian component pruning section, the truncation threshold of the Gaussian component is $10^{-5}$. The state extraction threshold was set to 0.5. The merging threshold was set to 10, the maximum number of Gaussian components was 100, the velocity threshold $M_{\mathrm{speed}}=10$, and the Euclidean distance $M_{d}=40$. The number of Monte Carlo simulations was 100. Evaluating tracking quality by OSPA distance:(49)$$OSPA_{p,c}\left(x_{k},{\hat{x}}_{k}\right)=\sqrt[{p}]{\frac{\min\sum_{i=1}^{\left|x_{k}\right|}{\left(d_{c}\left(x_{k}^{i},{\hat{x}}_{k}^{\pi \left(i\right)}\right)\right)}^{p}+c^{p}\left(\left|{\hat{x}}_{k}\right|-\left|x_{k}\right|\right)}{\left|{\hat{x}}_{k}\right|}}$$$x_{k}$ is the target state vector, the two parameters of OSPA distance were set to p=1 and c=200, respectively. The smaller the OSPA distance, the higher the accuracy of target state estimation.

### 4.2. Simulation Scenario 1

In this MTT scene, clutter rate λ=90, survival probability $p_{s,k}=0.99$, and detection probability $p_{D,k}=0.95$. Figure 1 is the real track and actual tracking measurement map of multiple targets in dense clutter and high detection probability environment. As can be seen from Figure 1b, in the MTT scene with dense clutter, the proposed improved GM-PHD algorithm can accurately obtain the target track. In Figure 1b, we can see that there are false tracks, which are caused by the clutter around the measured values of newborn targets. They will affect the stability of the filter before the track is formed. However, after clustering by speed and distance, the influence would gradually disappear with time, and the wrong target track would not appear.

#### 4.2.1. Filter Quality Analysis

Figure 2 compares the performance of various filters in dense clutter and high detection probability environment, including GM-PHD filter, label GM-PHD filter, robust label GM-PHD filter, and GM-CPHD filter. Obviously, the tracking performance of the labeled GM-PHD filter proposed in this paper was obviously better than GM-PHD, traditional labeled GM-PHD, and GM-CPHD in dense clutter environments. As can be seen from Figure 2a,b, the OSPA distance between GM-PHD and label GM-PHD filters is large, which shows that the dense clutter environment seriously affects the filter performance. As can be seen from Figure 2c,d, the improved labels GM-PHD and GM-CPHD have better performance in estimating the number of targets because the improved label GM-PHD filter combines Gaussian components with the same label and adds a threshold clustering separation step after merging and pruning, while GM-CPHD filter calculates the second moment of a posterior probability density function at the expense of operation time. Thus it is more accurate to estimate the number of multiple targets.

#### 4.2.2. Computational Complexity Analysis

In the process of pruning and merging the traditional GM-PHD filter and the label GM-PHD filter, all Gaussian components were pruned and merged. The algorithm proposed in this paper only processes the Gaussian components with the same label. It will undoubtedly greatly reduce the amount of calculation of the Gaussian component and avoid the problem of exponential increase in the amount of calculation. Moreover, the algorithm proposed in this paper also includes a threshold separation clustering method, which also processes Gaussian components with the same label. It can be seen in the following simulation that processing the Gaussian components with the same label will reduce the computational complexity very well.

Figure 3 compares the running time of different filters in this scenario. It can be seen that the improved label GM-PHD filtering algorithm improves computational efficiency. Compared with the GM-PHD filtering algorithm, it has slightly higher computational complexity but compared with the other two filtering algorithms, it has higher computational efficiency because it uses label sets to combine Gaussian components with the same label, thus reducing the computational complexity. Figure 3 also shows that the improvement of GM-CPHD tracking performance is at the expense of computing time to some extent. Compared with the GM-PHD filter, the operation time of the improved label GM-PHD filter is still 13.14% higher than that of the GM-PHD filter, but the computational efficiency is improved by 11.23% compared with the traditional label GM-PHD filter.

### 4.3. Simulation Scenario 2

In this MTT scene, clutter rate λ=90, survival probability $p_{s,k}=0.99$, and detection probability $p_{D,k}=0.8$. Figure 4 shows the tracking performance comparison of each filter in dense clutter and low detection probability environment. As can be seen from Figure 4, except for the improved label GM-PHD filtering algorithm, the tracking performance of other filtering algorithms is obviously degraded under the condition of low detection probability, which indirectly proves the robustness of the proposed algorithm. After pruning and merging, the Gaussian components are separated and clustered by using velocity and distance thresholds, which effectively improves the tracking performance of multi-targets under the environment of low detection probability.

### 4.4. Simulation Scenario 3

In this MTT scene, the survival probability $p_{s,k}=0.99$, detection probability $p_{D,k}=0.9$, the proposed algorithm under different clutter rate tracking performance superiority was verified by simulation. Figure 5 shows the performance comparison of algorithms with different clutter rates. As can be seen from Figure 5a–c, the tracking errors of the GM-PHD filter and the label GM-PHD filter increases linearly with the increasing clutter rate. The reason is that with the increasing clutter, the Gaussian component pruning and combining step becomes difficult to handle, resulting in error coupling of various errors. The improved label GM-PHD filtering algorithm uses the Gaussian component of the same label to process the subsequent steps, which effectively avoids these problems. It can be seen from Figure 5d that with the increase of clutter rate, the calculation amount of GM-CPHD increases linearly, while the calculation amount of the improved label GM-PHD filtering algorithm changes little, which also reflects the stability of the algorithm.

### 4.5. Simulation Scenario 4

In this MTT scene, the survival probability $p_{s,k}=0.99$ and clutter rate λ=60 were simulated to prove the performance advantages of the proposed algorithm in different detection probability environments. Figure 6 shows the comparison of algorithm performance under different detection probabilities. It can be seen from Figure 6 that the improved label GM-PHD filtering algorithm had higher stability than other algorithms and had certain advantages in tracking performance under different detection probabilities.

Simulations of the above four scenarios show that the improved label GM-PHD filtering algorithm has good tracking performance in dense clutter and low detection probability environment. It also has good stability in the environment where clutter rate and detection probability change.

This algorithm is based on GM-PHD and focuses on multi-target tracking in a dense clutter environment, and the motion of the target is also linear. In the next work, we can study non-linear high-maneuvering multi-targets and Multi-sensor observation scenes, using SMC algorithm for non-linear moving targets. By considering the tracking problems in the case of model uncertainty, the algorithms [16,17,18] can be more suitable for actual tracking scenarios.

## 5. Conclusions

Aiming at the problem of low accuracy and low computational efficiency of MTT in dense clutter environments, this paper proposes an efficient and accurate robust label GM-PHD filtering algorithm. Based on the traditional label GM-PHD, this algorithm improves the pruning and merging step to merge Gaussian components with the same label, which greatly reduces the computational burden and error merging in a dense clutter environment. According to velocity threshold and Euclidean distance, the newborn state extraction method can effectively reduce the occurrence of false tracks and improve the performance of MTT in dense clutter environments. The proposed algorithm can be further improved to adapt to MTT in an unknown parameter environment.

## Figures and Tables

**Figure 1 sensors-22-00070-f001:**
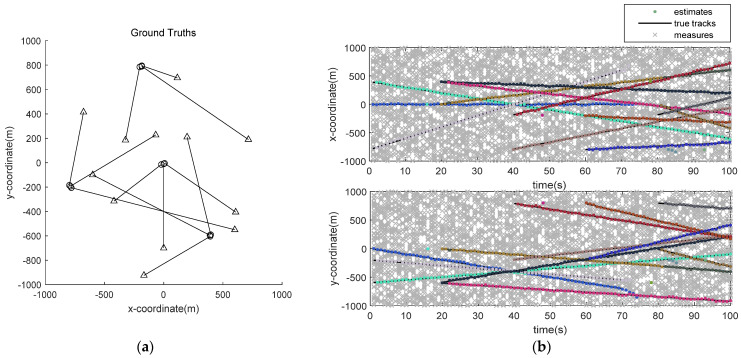
Multi-target true trajectory and actual tracking measurement: (**a**) True target trajectory; (**b**) Improved GM-PHD filter tracking results.

**Figure 2 sensors-22-00070-f002:**
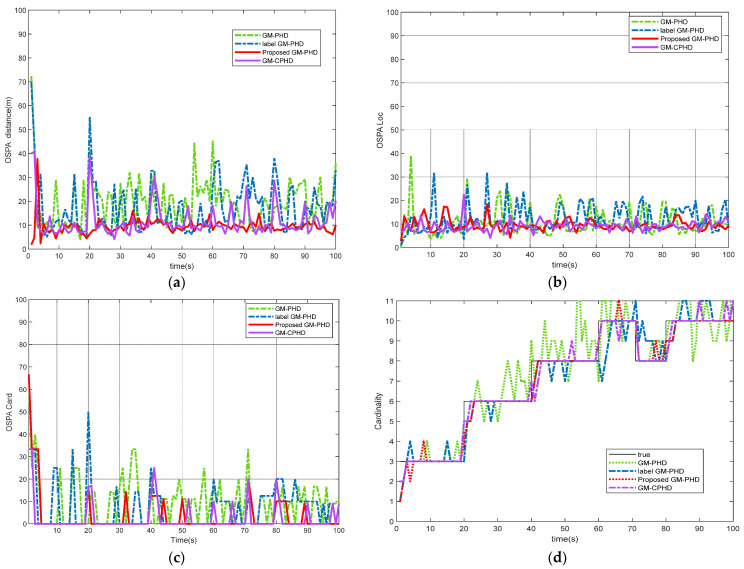
Performance comparison of different algorithms in dense clutter and high detection probability environment: (**a**) OSPA distance comparison; (**b**) OSPA Location comparison; (**c**) target cardinalities estimation error; (**d**) estimated cardinalities of targets.

**Figure 3 sensors-22-00070-f003:**
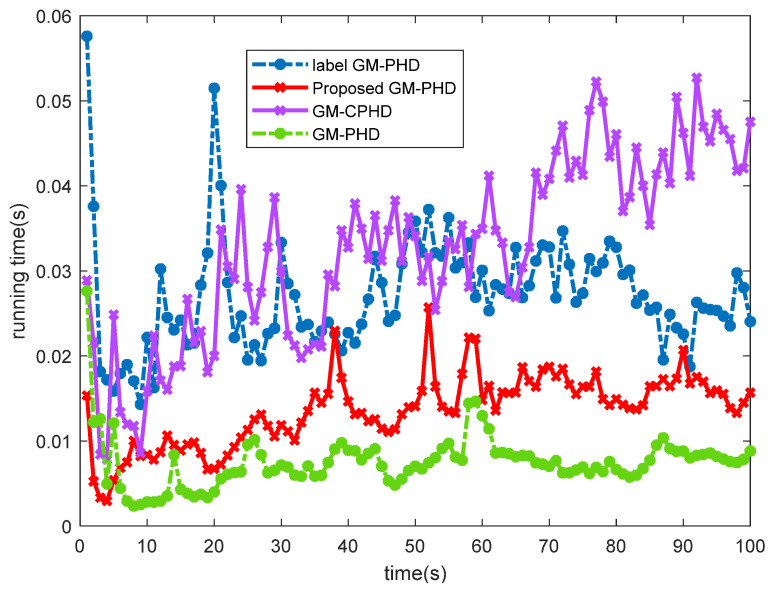
Different algorithm running time.

**Figure 4 sensors-22-00070-f004:**
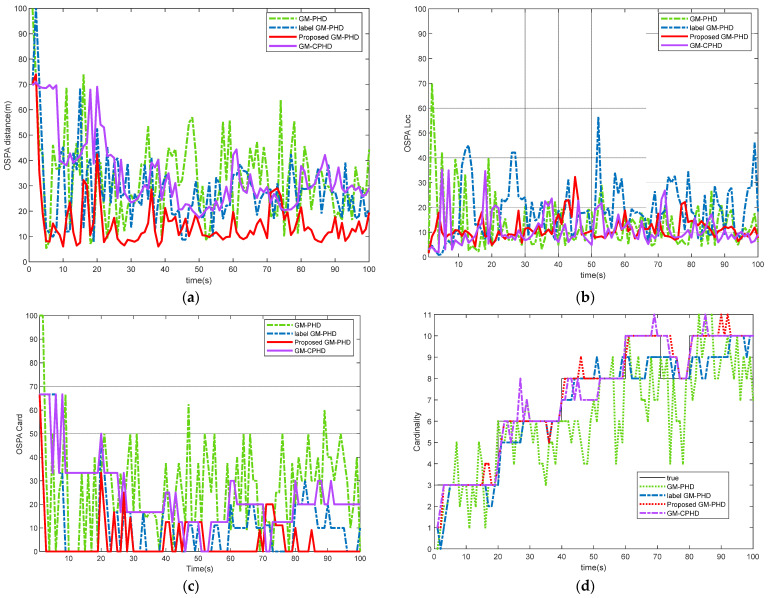
Performance comparison of different algorithms in dense clutter and low detection probability environment: (**a**) OSPA distance comparison; (**b**) OSPA Location comparison; (**c**) target cardinalities estimation error; (**d**) estimated cardinalities of targets.

**Figure 5 sensors-22-00070-f005:**
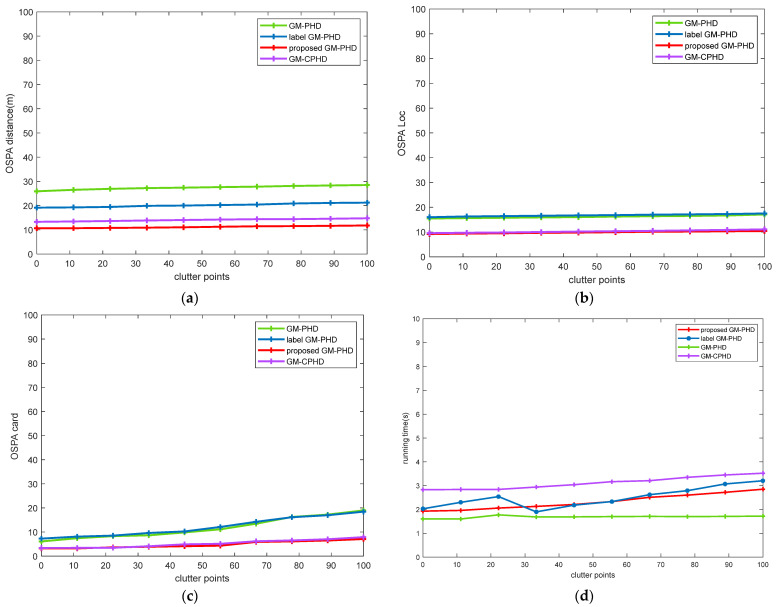
Comparison of the performance of different filters in a changing environment of clutter rate: (**a**) OSPA distance comparison; (**b**) OSPA Location comparison; (**c**) target cardinalities estimation error; (**d**) running time comparison.

**Figure 6 sensors-22-00070-f006:**
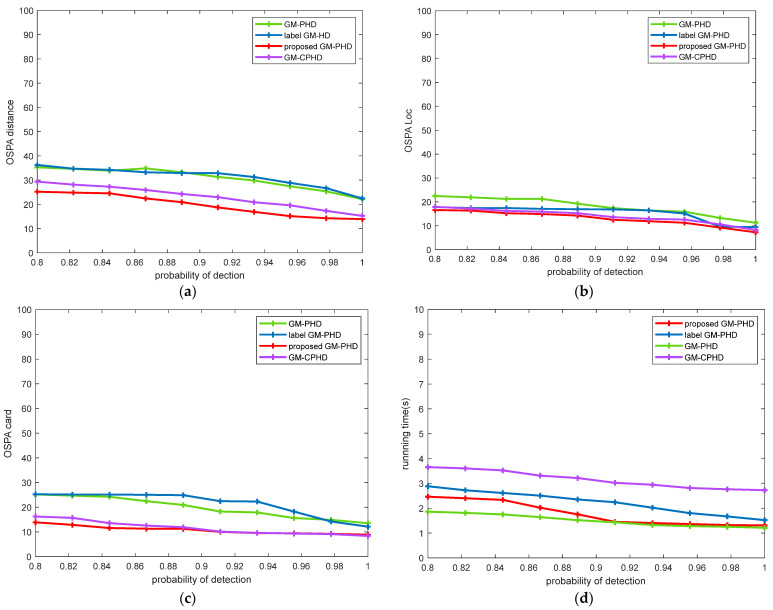
Comparison of the performance of different filters in a changing environment of detection probability: (**a**) OSPA distance comparison; (**b**) OSPA Location comparison; (**c**) target cardinalities estimation error; (**d**) running time comparison.

## Data Availability

Not applicable.

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
