# Peer review of "Label GM-PHD Filter Based on Threshold Separation Clustering"

_sensors, 2021, doi:10.3390/s22010070_

Round 1

Reviewer 1 Report

The paper contains interesting material and the application is important. The author studies a  new Gaussian mixture probability hypothesis density filter improving the performance of the competitors and allowing to produce continuous track in the tracking process. However, before a possible publication, some changes are required. Please following my suggestions below.

- The description of the techniques is quite poor and very short. At least, try to give all the details. For instance, in Section 2.1 please add a specific sentence regarding N_k saying if it is fix or not (I believe that not) and that you are inferring also N_k.

- Is J_k in Eq. (3) the same that N_k? please clarify.

Table is esthetically very ugly and  also provides no information. Please remove it.

- Please add a new section discussing the computational cost of the proposed filter.

- If you fix N_k (and that repeat the procedure for different N_k's) you can solve you problem with sophisticated particle filters able to do tracking and model selection at the same time (i.e., learning also static parameters). These filters can give also an importance weight to each N_k. Hence, you can do infer also regarding N_k; see for instance,

 C. C. Drovandi, J. McGree, and A. N. Pettitt. A sequential Monte Carlo algorithm to incorporate model uncertainty in Bayesian sequential design." Journal of Computational and Graphical Statistics, 23(1):324, 2014.

 L. Martino, J. Read, V. Elvira, F. Louzada, Cooperative Parallel Particle Filters for on-Line Model Selection and Applications to Urban Mobility, Digital Signal Processing Vol. 60, pp. 172-185, 2017.

 I. Urteaga, M. F. Bugallo, and P. M. Djuric. Sequential Monte Carlo methods under model uncertainty, IEEE Statistical Signal Processing Workshop (SSP), pages 15, 2016.

Please discuss this point (at least briefly). This discussion can improve the quality of the paper and the number of interested readers (hence, increase the impact of your work). This discussion can also clarify the degree of novelty of your work. 

Reviewer 2 Report

(1) The article is written clearly and has good readability.
(2) The literary contribution is very general. In view of the special issue, it can be considered to be published.

(3) Some contents of the manuscript are inspired by literature [1], and need to be cited, and the differences are pointed out.

(4) Some english grammatical errors.

[1] Luo Q ,  Gao Z ,  Xie C . Improved GM-PHD filter based on threshold separation clusterer for space-based starry-sky background weak point target tracking[J]. Digital Signal Processing, 2020, 103:102766.
